# Unveiling prognostics biomarkers of tyrosine metabolism reprogramming in liver cancer by cross-platform gene expression analyses

Tran N. Nguyen[ID][1]*, Ha Q. Nguyen[2], Duc-Hau Le[1]

1 Department of Computational Biomedicine, Vingroup Big Data Institute, Hanoi, Vietnam, 2 Department of Computer Vision, Vingroup Big Data Institute, Hanoi, Vietnam

* v.tranNN3@vinbdi.org

**Data Availability Statement:** All relevant data are within the manuscript and its Supporting Information files.

## Abstract

Tyrosine is mainly degraded in the liver by a series of enzymatic reactions. Abnormal expression of the tyrosine catabolic enzyme tyrosine aminotransferase (TAT) has been reported in patients with hepatocellular carcinoma (HCC). Despite this, aberration in tyrosine metabolism has not been investigated in cancer development. In this work, we conduct comprehensive cross-platform study to obtain foundation for discoveries of potential therapeutics and preventative biomarkers of HCC. We explore data from The Cancer Genome Atlas (TCGA), Gene Expression Omnibus (GEO), Gene Expression Profiling Interactive Analysis (GEPIA), Oncomine and Kaplan Meier plotter (KM plotter) and performed integrated analyses to evaluate the clinical significance and prognostic values of the tyrosine catabolic genes in HCC. We find that five tyrosine catabolic enzymes are downregulated in HCC compared to normal liver at mRNA and protein level. Moreover, low expression of these enzymes correlates with poorer survival in patients with HCC. Notably, we identify pathways and upstream regulators that might involve in tyrosine catabolic reprogramming and further drive HCC development. In total, our results underscore tyrosine metabolism alteration in HCC and lay foundation for incorporating these pathway components in therapeutics and preventative strategies.

## Introduction

Hepatocellular carcinoma (HCC) remains the most common cancer in the word, especially in Asia and Africa, and the third leading cause of cancer-related death worldwide [1]. It is believed that the pathogenesis of HCC is a long-term process that involves constant metabolic reprogramming. Previous efforts to investigate metabolic programming of HCC have largely focused on aerobic glycolysis, commonly referred to as the Warburg effect, which supports tumor growth in part by accumulating glycolytic intermediates for anabolic biosynthesis [2, 3]. For instance, HCC tumors express high levels of the hexokinase isoform 2 (HK2), which converts glucose to glucose-6-phosphate, and its expression is associated with the pathological stage of the tumor [4, 5]. HK2 silencing acted synergistically with sorafenib to inhibit HCC

**Funding:** This work was supported by Vingroup Big Data Institute. The funders had no role in study design, data collection and analysis, decision to publish, or preparation of the manuscript.

**Competing interests:** The authors have declared that no competing interests exist.

tumor growth in mice [5]. Besides glucose, HCC has been reported to alter its lipid and lipo-protein catabolic and anabolic pathways and increased HCC risks have been observed in patients with obesity [6], diabetes [7], and hepatic steatosis [8]. Recent studies defined a functional association among lipogenesis, multifunctional enzyme fatty acid synthase (FASN), sterol regulatory element-binding protein-1 (SREBP-1), a transcription factor regulating FASN expression, and HCC [9, 10].

Recently there are increasing evidences suggesting that cancer cells have increased levels of oxidative stress and ROS production compared to normal cells [11]. Thus, redox homeostasis which controls cell signaling and metabolism, is finely tuned in cancer cells [12, 13]. It is established that through the inhibition of PKM2 and subsequent metabolic switch, ROS allows cancer cells to tolerate responses to oxidative stress [12, 13]. Oxidative damage is considered as a key pathway in HCC progression and increases patient vulnerability for HCC recurrence [14]. As previously reported, accumulation of a *m*-tyrosine may disrupt cellular homeostasis and contribute to disease pathogenesis and the elimination of this isomer can be an effective defense against oxidative stress [15].

Tyrosine, like other amino acids, is the building block for proteins as well as an alternative energy source for cellular functions. Liver is the major organ where tyrosine degradation takes place to produce intermediates or precursors for gluconeogenesis and ketogenesis. The degradation of tyrosine is catalyzed through a series of five enzymatic reactions. Disturbed tyrosine metabolism has been implicated in several types of disease such as Huntington's disease [16] and esophageal cancer [17, 18]. Previously reported. patients with hereditary tyrosinemia are more likely to develop HCC [19, 20]. In patients with HCC, an upregulation of serum tyrosine has been recorded [21, 22], suggesting a deregulated tyrosine metabolism in HCC. However, to date, there is a lack of systematic study to profile the state of tyrosine catabolic enzymes and molecular impacts of alteration in tyrosine catabolism in HCC development.

As previously reported, the frequent deletion of 16q22 and aberrant methylation led to the downregulation of the first tyrosine catabolic enzyme TAT (tyrosine aminotransferase) [23]. Functional analyses showed that TAT harbored proapoptotic effect and that TAT suppression could promote liver tumorigenesis [23]. Glutathione S-transferases (GSTs) are a family of phase II isoenzymes that detoxify toxicant to lower toxic [24] and its dysfunction has been found to be closely related with response to chemotherapy [25–27]. *GSTZ1* belongs to the zeta class of GSTs and is the fourth enzyme in tyrosine metabolism. Patients carrying GSTZ1 variants had an increased risk of bladder cancer when exposed to trihalomethanes [28]. Furthermore, a computational-based investigation suggested *GSTZ1* might act as a protective factor in ovarian cancer [29].

In this study, we aim to systematically investigate the expression and prognostic value of tyrosine catabolism enzymes (TAT, HPD, HGD, GSTZ1 and FAH) in HCC by integrating large-scale datasets. We further detect enriched pathways associated with overexpression of a tyrosine catabolic enzyme in HCC cells. Our comprehensive, gene-centric analysis shed light on the genomic changes, clinical relevance, upstream regulators and possible impact of tyrosine catabolic genes on HCC development.

## Results

### A cross-platform, pan-cancer analysis of tyrosine catabolic enzyme expression

We first set out to investigate the expression profiles of tyrosine catabolic genes in cancer transcriptomes (Fig 1A). Here, we used the Oncomine online database [30] to perform pan-cancer transcriptome analysis on its available data sets. The top mRNA differences between cancer

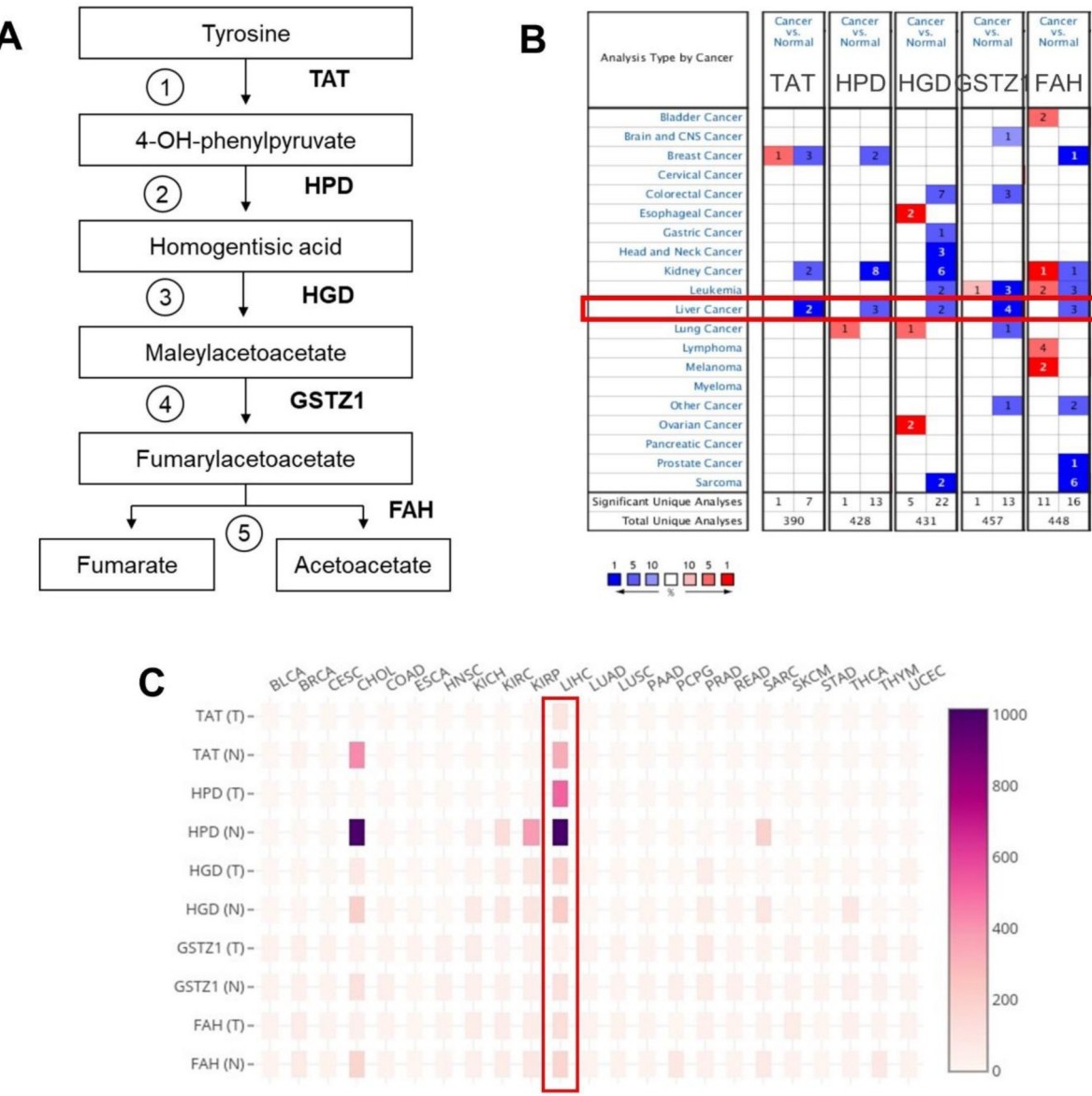

**Fig 1. Downregulation of the tyrosine catabolic genes in several types of cancer, including HCC.** (A) Graphics of tyrosine catabolism process. (B) The mRNA expression levels of the tyrosine catabolic genes according to Oncomine database. The mRNA expression of the genes (cancer versus normal tissue) in pan-cancers analyzed with the Oncomine database. The graphic demonstrates the numbers of datasets that meet our threshold in each cancer type. Cell color was defined as the gene rank percentile in the study. (C) The heat map indicates the expression after normalization by TPM+1 for comparison between tumor (T) and normal (N) across cancer types. Normal tissues are matched TCGA adjacent tissue and GTEx data. The cancer abbreviation names are shown according to TCGA study abbreviations (S1 Table). TPM, transcript per million.

samples and normal samples were analyzed by default selective criteria. Fig 1B showed that there was a total of 390, 428, 431, 457 and 448 Oncomine data sets involving the genes, *TAT*, *HPD*, *HGD*, *GSTZ1* and *FAH*, respectively (p ≤ 1e-04, fold-change threshold = 2). Remarkably, in most data sets, a large proportion of patients demonstrated downregulation of these genes in the tumorous parts compared to those of normal samples (red represents upregulation, blue represents downregulation). Specifically, in HCC, all of the gene sets show downregulation of the investigated tyrosine catabolic enzyme-encoding genes (Fig 1B, LIHC,

highlighted in red box). Furthermore, the Gene expression heat map from GEPIA pan-cancer transcriptome analysis [31] showed markedly downregulation of *TAT*, *HPD* and *GSTZ1* in HCC (Fig 1C). Additionally, in cervical squamous cell carcinoma (CESC), all of the tyrosine catabolic genes were visibly downregulated in tumors compared to normal tissue adjacent to the tumor (Fig 1C). Through this initial observation, we found evidences to support that tyrosine catabolic genes expression were downregulated in many cancers, including HCC.

## Tyrosine catabolic genes are downregulated in HCC

Next, to further investigate the role of tyrosine catabolic enzymes, we performed analysis of a publicly available dataset (The Cancer Genome Atlas [32] [TCGA], Liver Cancer [LIHC]) including gene expression in 369 HCC tissues and 160 normal liver tissues (including adjacent tissues and GTEx normal tissues). Here, the data demonstrated that *TAT*, *HPD* and *GTSZ1* were decreased in HCC tissues compared to normal liver (Fig 2A, cutoff $|Log_2FC| = 1$, cutoff p value = 0.01). However, the gene expression of *HGD* and *FAH* were virtually unchanged in HCC samples compare to normal liver samples.

To gain supporting evidence on the downregulation of tyrosine catabolic genes in HCC, the GSE89377 (Data Citation 1) dataset was employed to assess the expression of these genes in normal liver samples, early HCC and HCC from stage 1 to 3. Interestingly, we found that in early HCC, the expression of tyrosine catabolic genes was insignificantly changed compared to normal liver. However, the transcripts of *TAT*, *HPD*, *HGD*, *GSTZ1* and *FAH* significantly reduced in the HCC stage 2 and stage 3 compared to normal liver (Fig 2B, $p < 0.05$).

Overall, our combined analysis on TCGA data and an independent GSE dataset showed that tyrosine catabolic genes were downregulated in late stage HCC compared to normal liver.

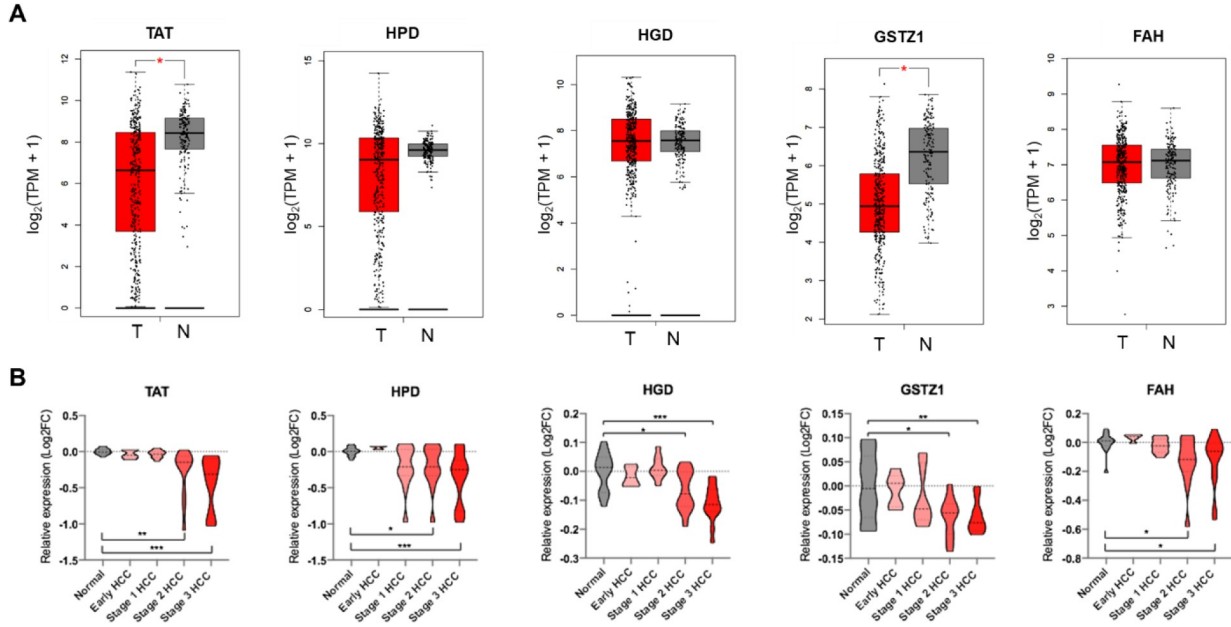

**Fig 2. Gene expression profile of the tyrosine catabolic genes in HCC.** (A) Gene expression analysis of tyrosine catabolic genes using GEPIA based on the TCGA and GTEx database. Box plots represent the gene expression level in terms of $log_2(TPM+1)$ in the tumor (red, n = 369) and normal (grey, n = 160) samples, respectively. Normal tissues are matched TCGA adjacent tissue and GTEx data. The method for differential analysis is one-way ANOVA. (B) Gene expression analysis across stages of the tyrosine catabolic genes in GSE89377 dataset. Violin plots represent $log_2(TPM+1)$ of genes in normal (grey, n = 13), early HCC (red, n = 5), stage 1 HCC (red, n = 9), stage 2 HCC (red, n = 12) and stage 3 HCC (red, n = 14). A t-test was used to compare the expression difference between tumor and normal tissue; $p < 0.05$ was considered statistically significant. $^*p < 0.05$, $^{**}p < 0.01$, $^{***}p < 0.001$ based on the Student's t test. Values are mean ± SEM. TPM, transcript per million.

## Prognostic value of tyrosine catabolic genes in patients with HCC

Subsequently, we sought to determine the clinical relevance of *TAT*, *HPD*, *HGD*, *GSTZ1* and *FAH* expression in term of prognosis in HCC patients since these genes were highly enriched in liver tissues (S1 Fig). Kaplan–Meier analysis was employed to compare between the sub-groups with high and low gene expression (using the median, 25% or 75% quartile values of gene expression as cut-off points) in TCGA-LIHC cohort of 364 liver cancer patients. The overall survival was significantly associated with *TAT*, *HGD* and *GSTZ1* expression in HCC samples (p = 0.0067, p = 0.0039 and p = 0.036, respectively) (Fig 3). Similarly, lower expression of *TAT*, *HGD* and *GSTZ1* could also translate to a worse disease-free survival in HCC patients (p = 0.011, p = 0.0038 and p = 0.036, respectively) (S2 Fig).

To further validate the potential application of tyrosine catabolic genes in the clinic, we extracted the characterized IHC images from the Human Protein Atlas. HCC tumor tissue staining of tyrosine catabolic enzymes showed significant decrease in positive staining compared with normal liver tissue. Specifically, HPD staining decreased by 2.26-fold ± 2.10 (p = 0.0388), HGD decreased by 1.67-fold ±0.87 (p = 0.0423) and GSTZ1 decreased by 2.27-fold ±1.09 (p = 0.0007) in HCC tumor compared to normal liver tissue (Fig 4).

These findings highlighted that the expression of tyrosine catabolic enzyme-encoding genes correlated with worse overall survival and disease-free survival in HCC and that TAT, HGD and GSTZ1 had potential prognostic value in patients with HCC.

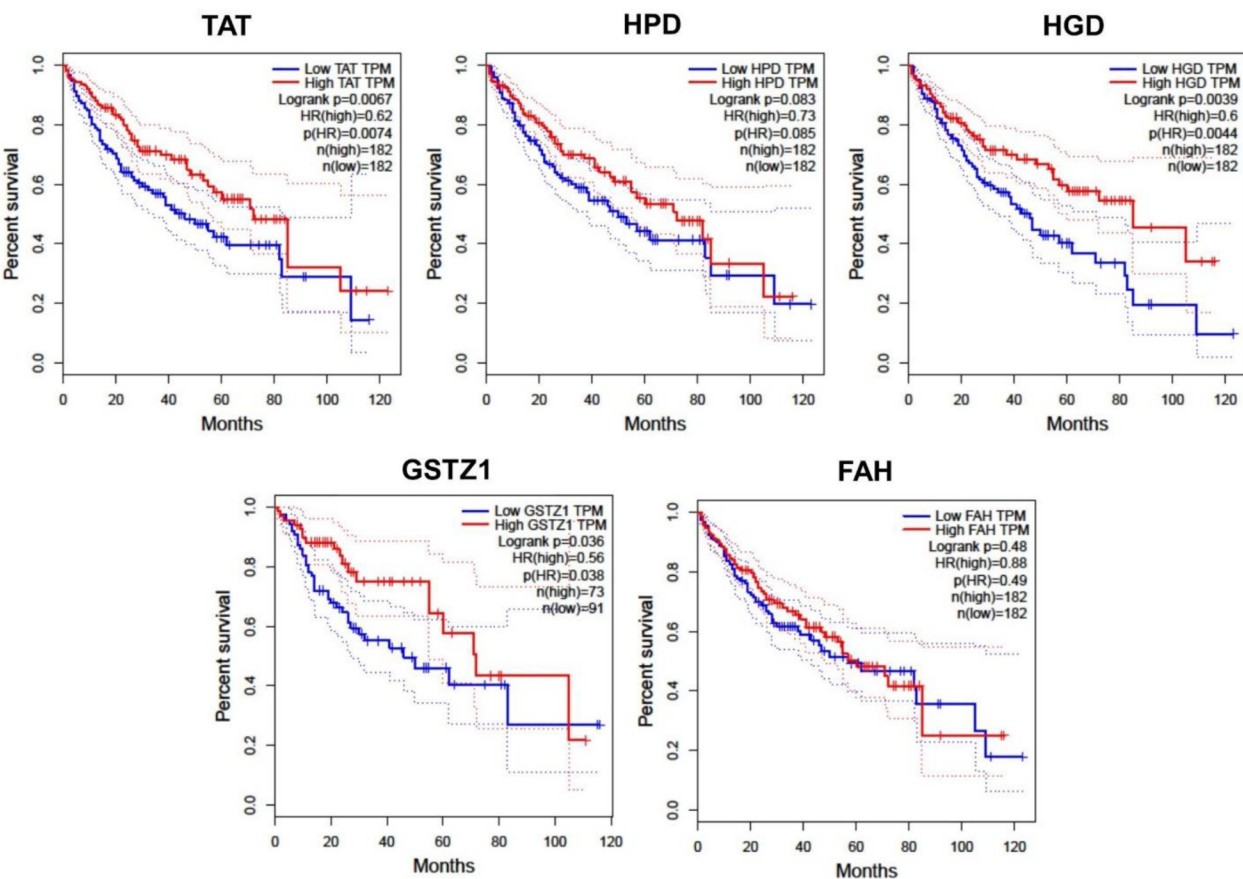

**Fig 3. Overall survival outcomes of HCC patients.** Data on from 364 patients were analyzed using log-rank tests based on gene expression in HCC tissues from the TCGA cohort. Kaplan-Meier curves are plotted using GEPIA for TAT, HPD, HGD, GSTZ1 and FAH, and HRs and 95% confidence intervals are shown. Abbreviation: HCC, hepatocellular carcinoma, HRs, hazard ratios; TCGA, the Cancer Genome Atlas.

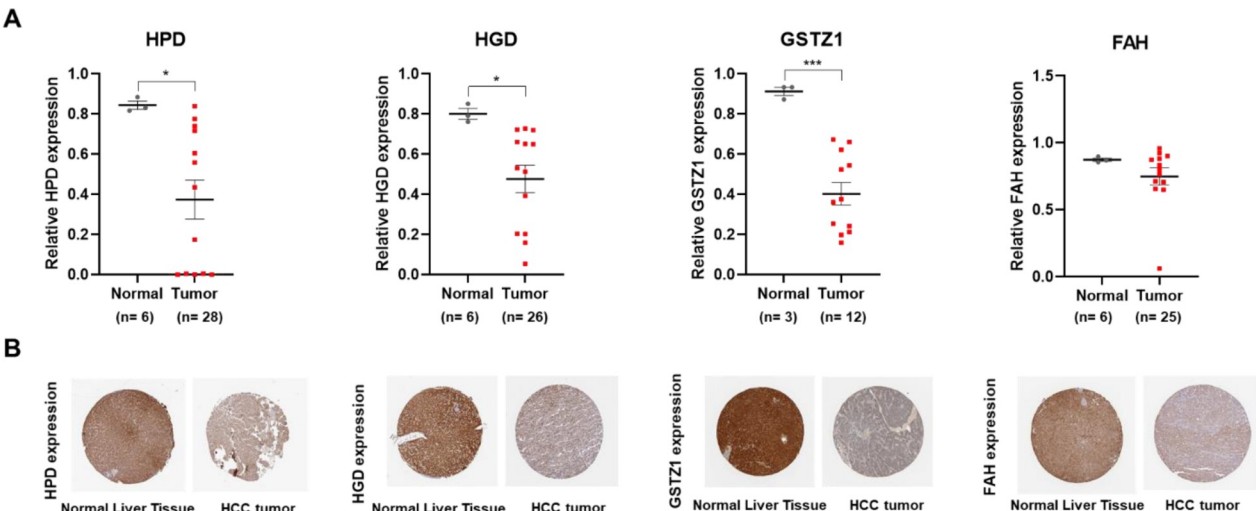

**Fig 4. The protein expression profile of the tyrosine catabolic genes in the pan-cancer analysis.** (A) Quantification of HPD, HGD, GSTZ1 and FAH expression in IHC images obtained from HPA. A t-test was used to compare the expression difference between tumor and normal tissue adjacent to the tumor; $p < 0.05$ was considered statistically significant. *$p < 0.05$, ***$p < 0.001$ based on the Student's t test. Values are mean ± SEM. (B) Representative images of normal liver tissue and HCC tissue stained with antibody against the enzymes.

## Gene expression profiling of GSTZ1 expressing HCC cell line

Following the previous analyses, we noted that the fourth rate-limiting enzyme, GSTZ1 (Gluta-thione S-transferase Zeta 1) had significant downregulation and prognosis. We therefore sought to study the molecular pathway alterations associated with this gene. Here, we explored the publicly available data set GSE117822 (Data Citation 2) where GSTZ1 is overexpressed in Huh7 HCC cell line by adenoviral transfection. R software [33] with the DESeq2 [34] package was applied to screen DEGs from the gene expression dataset GSE between control vectors and overexpressed GSTZ1. A total of 3163 DEGs (p <0.01) were identified from this dataset, 1742 upregulated genes and 1421 downregulated genes.

To investigate changes in molecular pathways associated with GSTZ1 overexpression, we use GSEA to rank the DEGs against the C2 canonical pathway gene set [35]. We were able to profile positively and negatively enriched pathways in GSTZ1 overexpressed Huh7 (S3 Fig). For better visualization of related gene sets and identification of important pathway families, we presented the pathways using Enrichment Map [36] in Cytoscape [37] (Fig 5). As expected, we observed a positive enrichment for multiple metabolism related pathways including Metab-olism of Lipids, Metabolism of Proteins and Metabolism of Amino Acids. Noticeably, increased GSTZ1 expression led to heightened Oxidative Phosphorylation and Respiratory Electron Transport. Previously published in hepatocytes, limited oxidative phosphorylation activity associated with decreased apoptotic cell-death and increased cancer development [38]. On the other hand, genes involved in glycolysis, such as *HK2*, *PDK2* were downregulated (1.88-fold and 2.05-fold, respectively) in cell expressing GSTZ1 compared with vector control. It is known that the glycolytic gene *HK2* is highly expressed in HCC [39]. In cirrhosis, increased expression of glycolytic genes associated with higher HCC risk [40]. Most impor-tantly, overexpression of GSTZ1 in HCC cell led to the downregulation in several pathways in cancer gene sets (Kegg Small Cell Lung Cancer and Kegg Chronic Myeloid Leukemia). Together, these data highlighted the changes in molecular pathways that correspond to GSTZ1 expression and critically, provided insights on how overexpression of GSTZ1 might negate HCC development.

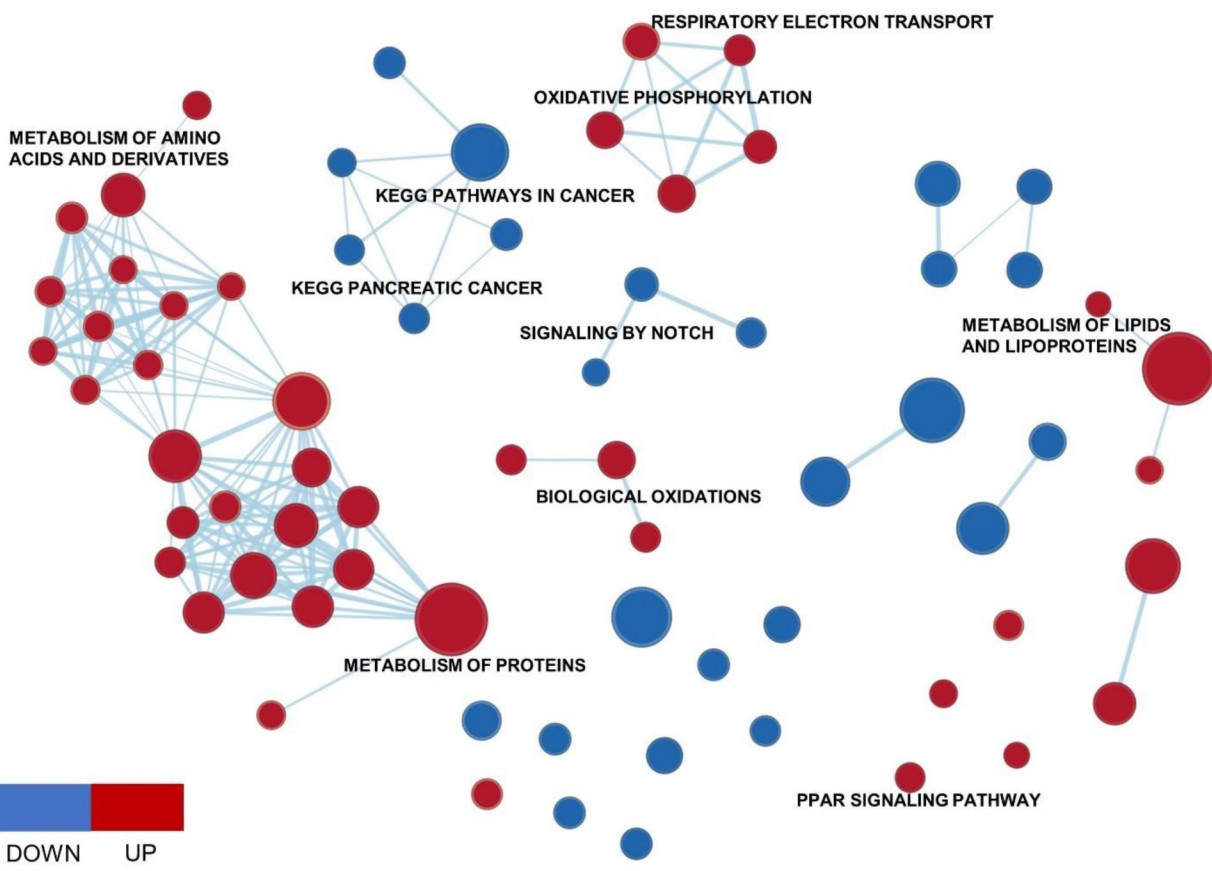

**Fig 5. Enrichment Map of GSTZ1 overexpressed huh7 and non-targeted control.** GSEA was used to obtain canonical pathway gene sets that were visualized using the Enrichment Map plug-in for Cytoscape. Each node represents a gene set with similar nodes clustered together and connected by edges with the number of known interactors between the nodes being represented by the thickness of edges. The size of each node denotes the gene set size for each specific.

## Mutation profiles of tyrosine catabolic genes in HCC

We extended our studies to investigate on underlying mechanism of how tyrosine catabolic genes were downregulated in HCC. First, we explored mutation profiles of tyrosine catabolic genes in 353 HCC patients by exploring TCGA data using cBioPortal [41]. We found that each individual gene was mutated in less than 1.1% of patients with HCC (S4 Fig). In all genes, there were 8/21 missense mutations that harbor deleterious effect (Table 1). However, when incorporating mutation type with mRNA expression profile, we did not observe a correlation where amplification led to increased expression or vice versa (S5 Fig). Second, we explored copy number status of tyrosine catabolic genes using data from GISTIC analysis [42] and cBioPortal (http://cbioportal.org). We found that even though the genes were located near the peak region of deletion, none of them were in focal (Table 2). Except for *HPD* (Q value = 0.019), the rest of the genes were less likely to suffer copy number alterations (Table 2). Taken together, we found that several base substitution mutation scenarios can lead to the deletion of tyrosine catabolic genes but these is not a strong association between mutation status and mRNA expression.

## MicroRNAs regulate the expression of tyrosine catabolic genes

Next, we sought out to explore microRNAs as possible negative regulators of *TAT*, *HPD*, *HGD*, *GSTZ1* and *FAH*. Using Target Scan database [43], we found there were two

**Table 1. Summary of mutations of tyrosine catabolic genes in patients with HCC.**

| Gene | DNA change | Type | Consequences | SIFT Impact |
|---|---|---|---|---|
| TAT | chr16:g.71568080 G>C | Substitution | 3 Prime UTR | N/A |
| | chr16:g.71570753_71570754insG | Insertion | Frameshift | N/A |
| | chr16:g.71576063G>T | Substitution | Intron | N/A |
| | chr16:g.71568109A>T | Substitution | 3 Prime UTR | N/A |
| | chr16:g.71570812T>C | Substitution | Missense | Deleterious |
| | chr16:g.71572596A>G | Substitution | Synonymous | N/A |
| | chr16:g.71568283C>A | Substitution | Missense | Deleterious |
| HPD | chr12:g.121847089T>C | Substitution | Missense | Deleterious |
| | chr12:g.121849748T>C | Substitution | Missense | Deleterious |
| | chr12:g.121858824G>A | Substitution | 5 Prime UTR | N/A |
| HGD | chr3:g.120646351T>A | Substitution | Missense | Deleterious |
| | chr3:g.120650834A>T | Substitution | Missense | Deleterious |
| | chr3:g.120670454C>A | Substitution | Missense | Deleterious |
| | chr3:g.120682178delTTCT | Deletion | 5 Prime UTR | N/A |
| GSTZ1 | chr14:g.77330329G>T | Substitution | Missense | N/A |
| FAH | chr15:g.80172237A>G | Substitution | Missense | Deleterious |
| | chr15:g.80160464G>T | Substitution | Splice Region | N/A |
| | chr15:g.80173063G>A | Substitution | Synonymous | N/A |
| | chr15:g.80186294G>T | Substitution | 3 Prime UTR | N/A |
| | chr15:g.80186299G>A | Substitution | 3 Prime UTR | N/A |
| | chr15:g.80162464C>T | Substitution | Intron | N/A |

microRNAs that targeted *TAT*, *HPD*, *GSTZ1* and *FAH* (Fig 6A), which were miR-539 and miR-661. There were no common microRNAs that target all tyrosine catabolic genes. First, investigation of 370 HCC samples and 50 normal samples (TCGA-LIHC) showed that miR-539 increased by 2.84-fold in HCC samples compared to normal liver (p = 0.05). Second, pan-cancer co-expression analysis for miRNA-target interaction in HCC using starBase [44] showed that miR-539 level negatively correlated with TAT, HPD, GSTZ1 and FAH expression ($r$ = -0.221, $r$ = -0.193, $r$ = -0.123, $r$ = -0.166) (S6 Fig). More importantly, our Kaplan-Meier analysis by KM-plotter [45] of TCGA-LIHC data set showed that high miR-539 expression led to worse overall survival in in HCC patients (Fig 6B).

Additionally, Kaplan-Meier analysis on CapitalBio miRNA Array liver dataset [47] also showed that miR-661 expression positively correlated with worse overall survival (Fig 6C). Overall, these findings suggested that in HCC, the downregulation of tyrosine catabolic genes can be due to microRNA regulation. We found that miR-539 and miR-661 can potentially suppress TAT, HPD, GSTZ1 and FAH expression and that expression of miR-539 and miR-661 can provide prognostic insights for patients with HCC.

**Table 2. Summary of CNAs of tyrosine catabolic genes in patients with HCC.**

| Gene name | Location | Nearest peak | In peak? | Q-value | Frequency of detection | | |
|---|---|---|---|---|---|---|---|
| | | | | | Overall | Focal | High value |
| TAT | chr16:71600753–71610998 | chr16:78129906–79627535 | No | 1 | 0.4108 | 0.0135 | 0 |
| HPD | chr12:122277432–122326517 | chr12:123453469–133155338 | No | 0.0191 | 0.1432 | 0.0486 | 0 |
| HGD | chr3:120347014–120401418 | chr3:114042610–115341566 | No | 1 | 0.1135 | 0.0081 | 0 |
| GSTZ1 | chr14:77787229–77797940 | chr14:66969095–67653632 | No | 0.856 | 0.3405 | 0.0324 | 0.0054 |
| FAH | chr15:80445232–80478924 | chr15:88785838–101883952 | No | 1 | 0.1892 | 0.0189 | 0 |

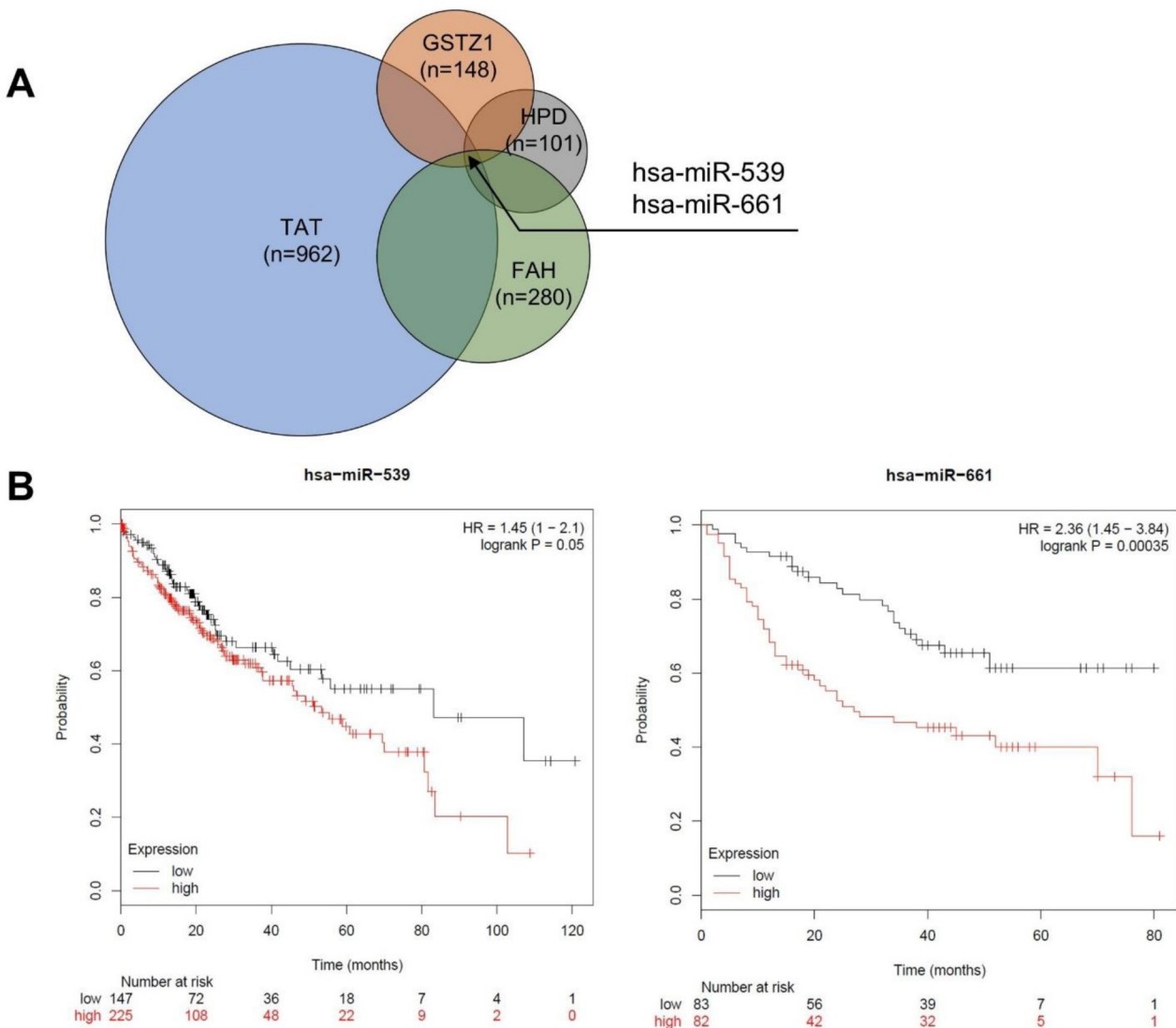

**Fig 6. Prognostic value of microRNAs that target the tyrosine catabolic genes.** (A) The Venn diagram demonstrated the number of predicted miRNAs that target TAT, HPD, FAH and GSTZ1 from TargetScan database. (B) Survival analysis with miR-539 and miR-661 (KM Plotter dataset). The TCGA-LIHC dataset [46] from Kaplan-Meier Plotter [60] was used to test for survival prediction capacity of miR-539 in liver cancer. The CapitalBio miRNA Array liver dataset [47] was used to test for survival prediction capacity of miR-661 in liver cancer. Cox regression model was used for each gene to predict relapse-free survival. Samples are divided into Low (black) and High (red) expression groups for each gene. Hazard ratio (HR) and p value for each association are shown within each plot.

## Discussion

We explored publicly available gene expression data sets and database to investigate the roles of genes in the tyrosine degradation pathway in the development of HCC. Our exploration indicated that all tyrosine catabolic genes decreased in HCC compared to normal liver tissues. We learned that the fourth rate-limiting enzyme, GSTZ1 expression significantly reduced, either in protein or mRNA level, in HCC (Figs 2A and 4). Even though the tyrosine catabolic gene expression remained unchanged at early stage HCC, they were significantly down-regulated in late stage HCC (Fig 2B). We also found that TAT, HGD and GSTZ1 expression levels

positively correlated with overall survival and disease-free survival of HCC (Figs 3 and S2). Previously shown, TAT, the first rate-limiting enzymes in the pathways, was downregulated in HCC [23, 48]. Functional *in vitro* validations showed that TAT induced apoptosis and that TAT possessed tumor-suppressive functions [23].

GSTZ1, which is expressed in both hepatic cytosol and mitochondria, has shown to be oxidative stress-related [29]. High levels of GSTZ1 expression conferred resistance to the effect of anti-cancer therapy of dichloroacetate in hepatocellular carcinoma cell lines by an independent mechanism to tyrosine metabolism [49, 50]. We explored a public dataset where GSTZ1 were overexpressed in HCC cell line Huh7 and found that with the expression of GSTZ1, there was positive enrichment of oxidative phosphorylation (Figs 5 and S2). It was published that restrained oxidative phosphorylation activity has been shown to favor hepatocarcinogenesis *in vivo* [38]. Additionally, we detected an overall enrichment in metabolism of proteins and lipids pathways and decrease in glycolysis genes following GSTZ1 expression (Fig 5). Liver is a dynamic organ which constantly undergoes metabolic shift. Cancer cells, including HCC, usually switch to aerobic glycolysis to maximize energy usage and further fuel growth [2]. Since overexpression of GSTZ1 associated with downregulation of several glycolytic genes, we consider it possible that the suppression of tyrosine catabolism can be a mechanism by which HCC switch to aerobic glycolysis during cancer progression.

The downregulation of other genes in the tyrosine catabolic pathways have not been linked to changes in DNA. Thus, we reason that the downregulation of *HPD*, *HGD*, *GSTZ1* and *FAH* might be dependent or independent of the downregulation of TAT. We found that four out of five genes were predicted to be regulated by miR-539, miR-661. Noticeably, miR-539 significantly increased in HCC compared to normal skin and that the miR-539 level inversely correlated with expression of *TAT*, *GSTZ1*, *HPD*, *FAH* (S4 Fig). In HCC patients, expression of two of these microRNAs positively correlated with overall survival (Fig 6B). Previously reported, miR-539 was usually downregulated and acted as tumor suppressors in various tumor types [51, 52]. In HCC, miR-539 was also demonstrated to suppress HCC development *in vitro* by targeting FSCN1 and suppressing apoptosis [53, 54]. Here, our findings suggested that on miR-539 might be a tumor promoter in contrast to previous experimental studies. On the other hand, prior studies showed that miR-661 was a tumor promoter in non-small cell lung cancer, colon cancer and ovarian cancer [55–57]. However, the roles of miR-661 in HCC development has not been investigated. Taken together, we speculate that miR-539 and miR-661 can be use as potential HCC prognosis markers and can serve as targets of interest for many functional studies.

Tyrosine metabolism is an important process that is often dysregulated in various diseases including cancers and chronic disorders [58]. Tyrosinemia type I patients have a higher risk of developing HCC [58]. The reasons for this high incidence of HCC are unknown [58]. A metabolomics study on esophageal cancer (EC) showed that tyrosine decreased in serum of patients with EC compared with healthy control [59, 60]. There has been little evidence on how tyrosine metabolism might contribute to cancer development even though changes in expression of some tyrosine metabolic genes have been reported in HCC patients [23, 61]. Our interesting findings add to existing body of knowledge in tyrosine catabolism in HCC.

## Conclusions

Our results from the integrative databases and comprehensive analysis of this study demonstrated the downregulation of tyrosine catabolic genes and their prognostic value in HCC (Figs 1–4 and S1 and S2). We provided evidence on how expressing these genes in HCC can negate HCC development (Figs 5 and S3) as well as identified candidate microRNAs that can regulate

the expression of tyrosine catabolic genes, which might be used as potential prognostic biomarker for HCC (Figs 6 and S6). Even though this study provided an interesting observation of tyrosine catabolic genes in HCC and data to support, further *in vitro* and *in vivo* experiments need to be applied to reveal the mechanism through which tyrosine catabolism affects in HCC development.

# Materials and methods

## Oncomine analysis

The Oncomine online databases [30] were accessed for the visualization of gene expression. Oncomine is an online cancer microarray database used to facilitate and promote discoveries from genome-wide expression analyses. The pan-cancer studies in Oncomine were selected to compare the expression levels in tumor vs normal tissue adjacent to the tumor. The selection criteria for the Oncomine studies were $p < 0.05$ as a threshold, 2-fold change and gene rank in the top 10%. The p value, fold changes, and cancer subtypes were extracted.

## Gene expression analysis

The TCGA data was analysed by GEPIA [31] (http://gepia.cancer-pku.cn/). For the differential expression analysis, the genes were $\log_2(TPM + 1)$ transformed. One-way ANNOVA was used to compute p value. Those with $\log_2(TPM+1) > 1$ and $p < 0.01$ were then considered differentially expressed genes. Normal tissues are matched TCGA adjacent tissue and GTEx normal tissue.

## Survival analysis

The overall survival curves of *TAT*, *HPD*, *HGD*, *GSTZ1* and *FAH* were investigated using the Kaplan-Meier method with the log-rank test. We set the high and low gene expression level groups by the median value for TAT, HPD HGD and FAH. For GSTZ1, the cutoff for high expression group is 75% and cutoff for low expression group is 25%. The overall survival plot was obtained with the hazards ratio (HR, based on Cox PH Model), the 95% confidence interval information and the p value. The log-rank p value was calculated with $< 0.05$ considered statistically significant. The whole process was implemented using the web-based tool GEPIA [31].

The prognostic values of hsa-miR-539 and has-miR-661 in HCC were analyzed using Kaplan Meier plotter (KM plotter) database [62]. Survival data of has-miR-539 was derived from the "RNA-seq" dataset which has 421 samples. Survival data of has-miR-661 was derived from the "Non-commercial spotted" dataset which has 166 samples. In brief, the miRNAs were entered into the database, the best cutoff is automatically selected, after which survival plots were generated and hazard ratio, 95% confidence intervals, log rank p value were displayed on the webpage. The log rank p value was calculated with $< 0.05$ considered statistically significant.

## Gene expression omnibus data mining

We retrieved transcriptome profiles of HCC tissues from GEO which is a public genomics database, allowing users to investigate gene expression profiles of interest [63]. The GSE89377 is a microarray dataset of multi-stage HCC in a GPL570 Affymetrix Human Genome U133 Plus 2.0 Array Platform. It contains 108 samples in total, including 13 healthy people, 5 with early HCC, 9 with Stage 1 HCC, 12 with Stage 2 HCC and 14 with Stage 3 HCC.

Processed gene expression dataset was downloaded using *GEOquery* [64]. *Limma* [65] R packages was used to determine the DEGs between normal and HCC tissues. $p \leq 0.01$ was used as the cutoff value.

### Differentially expressed genes identification and GSEA

The GSE117822 dataset was processed by Bioconductor package DESeq2 [34] to identify DEGs (fold change cut-off $\geq 1$ and significance p value $\leq 0.01$) and analyzed by GSEA with the Molecular Signatures Database "Canonical Pathways" gene set collection [35]. The default GSEA basic parameters were used.

### Quantification of immunohistochemistry images from Human Protein Atlas

Immunohistochemistry (IHC) images were downloaded from the publicly available The Human Protein Atlas [66] (HPA; http://www.proteinatlas.org) version 8.0. The analyses in present study were performed using HPA images of liver sections that were labeled with antibodies for HPD (antibody HPA038321), HGD (antibody HPA047374), GSTZ1 (antibody HPA004701) and FAH (antibody HPA041370). A custom script written in MATLAB programming language was used to detect positive staining based on brown pixel-counting. The absolute amount of antibody-specific chromogen per pixel was determined and normalized against total tissue area. Code is available at http://github.com/nguyenquyha/IHC-method. Statistical analyses were performed using GraphPad Prism 8.0.2. Unpaired student's t test was used. $P \leq 0.05$ was considered statistical significant.

### Identify miRNA candidates by Targetscan

Targetscan [67] database (http://www.targetscan.org) were accessed for identifying miRNA candidates. In brief, gene name was entered to retrieve a list of microRNAs that was predicted to target the input gene. Predicted microRNAs with non-canonical site type were not considered. After that, the predicted miRNA lists were compared to find common miRNAs that target *TAT*, *HPD*, *GSTZ1* and *FAH*.

### Copy Number analysis

Copy number alteration data from Gene-Centric GISTIC analyses was retrieved from TCGA Copy Number Portal (http://portals.broadinstitute.org/tcga/home). Liver hepatocarcinoma tumor type was selected for this analysis using the stddata__2015_04_02 TCGA/GDAC tumor sample sets from FireHose. In brief, after the analysis version was chosen and gene names were entered into the GISTIC browser, deletions data was generated with the default parameters (deletion threshold = 0.1, broad length cutoff = 0.5, peak confidence level = 0.95, significance threshold = 0.25). Gene copy number was plotted against gene expression level using cBioportal.

### Statistical analysis

Statistical analyses were performed using GraphPad Prism 8.0.2. Independent Student's t test was used to compare the mean value of two groups. Bars and error represent mean ± standard deviations (SD) of replicate measurements. Statistical significance was defined as $p \leq 0.05$. $*p \leq 0.05$, $**p \leq 0.01$ and $***p \leq 0.001$.

**Resource table.**

| Software and Algorithms | Version | Source |
|---|---|---|
| GraphPad PRISM | 8.0.2 | https://www.graphpad.com |
| R | 3.5.3 | https://www.r-project.org/ |
| limma R package | 3.8 | https://bioconductor.org/packages/release/bioc/html/limma.html |
| Cytoscape | 3.7.1 | https://cytoscape.org/ |
| EnrichmentMAP | 3.2.0 | http://apps.cytoscape.org/apps/enrichmentmap |
| GEPIA | 1 | http://gepia.cancer-pku.cn |
| Oncomine | NA | https://www.oncomine.org |
| KMPlotter | NA | https://kmplot.com |
| GSEA software | 2–2.2.3 | http://software.broadinstitute.org/gsea/index.jsp |

## Supporting information

**S1 Fig. Tyrosine catabolism enzyme-encoding gene expression across TCGA pan-cancer datasets.** Plots were taken from GEPIA online databases (http://gepia.cancer-pku.cn). Data indicates expression after normalization by $\log_2(TMP+1)$ for comparison between tumor and normal tissues in pan-cancer. The cancer abbreviation names are shown according to TCGA study abbreviations.
(DOCX)

**S2 Fig. Disease-free survival outcomes of HCC patients.** Disease-free survival outcomes of 364 HCC patients were analyzed using log-rank tests based on gene expression in HCC tissues from the TCGA cohort. Kaplan-Meier curves are plotted using GEPIA for TAT, HPD, HGD, GSTZ1 and FAH, and HRs and 95% confidence intervals are shown. Abbreviation: HCC, hepatocellular carcinoma, HRs, hazard ratios; TCGA, the Cancer Genome Atlas.
(DOCX)

**S3 Fig. Enriched GSEA canonical pathways of differentially expressed genes in GSTZ1 overexpressed liver cancer cells.** GSEA of canonical pathways for differentially expressed genes in GSTZ1 overexpressed liver cancer cells compared to empty vector control. The top twenty significantly enriched canonical pathways (both upregulated and downregulated) were displayed with their corresponding normalized enrichment score. Multiple pathways appear to be related to metabolism, oxidations and cancer development.
(DOCX)

**S4 Fig. The mutation profiles of tyrosine catabolic genes.** The mutation profiles of TAT, HPD, HGD, GSTZ1 and FAH was obtained from cBioPortal (Liver hepatocellular Carcinoma, TCGA, PanCancer Atlas, 353 samples).
(DOCX)

**S5 Fig. Gene copy number relative to gene expression of tyrosine catabolic genes.** Comparisons of TAT, HPD, HGD, GSTZ1 and FAH relative mRNA expression levels to putative mutation types and relative copy number. Data and plots were obtained from cBioPortal (Liver hepatocellular Carcinoma, TCGA, PanCancer Atlas, 353 samples).
(DOCX)

**S6 Fig. The anti-correlation between miR-539 and tyrosine catabolic genes.** The anti-correlation (Pearson correlation: r<0, p<0.05) between miR-539 and TAT, HPD, HGD and GSTZ1 in hepatocellular carcinoma was obtained from starBase[35] co-expression analysis on

TCGA-LIHC dataset.
(DOCX)

**S1 Table. TCGA study abbreviation.**
(DOCX)

## Acknowledgments

We also thank Dr. Ban Xuan Dong at Institute for Molecular Engineering, University of Chicago for proof-reading the manuscript.

## Author Contributions

**Conceptualization:** Tran N. Nguyen.

**Data curation:** Tran N. Nguyen.

**Formal analysis:** Tran N. Nguyen, Ha Q. Nguyen.

**Investigation:** Tran N. Nguyen.

**Methodology:** Tran N. Nguyen, Ha Q. Nguyen, Duc-Hau Le.

**Project administration:** Tran N. Nguyen.

**Supervision:** Tran N. Nguyen.

**Validation:** Tran N. Nguyen.

**Visualization:** Tran N. Nguyen.

**Writing – original draft:** Tran N. Nguyen.

**Writing – review & editing:** Tran N. Nguyen, Duc-Hau Le.

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
