## [Decision Letter · Decision Letter 0]

21 Apr 2020

PONE-D-20-02954

Unveiling Prognostics Biomarkers of Tyrosine Metabolism Reprogramming in Liver Cancer by Cross-platform Gene Expression Analyses

PLOS ONE

Dear Dr. Nguyen TN,

Thank you for submitting your manuscript to PLOS ONE. After careful consideration, I feel that it has merit but does not fully meet PLOS ONE’s publication criteria as it currently stands. Although the reviewers found the study interesting, they raised a number of serious points that do not allow me to accept the manuscript on the basis of how it is currently presented. Therefore, I invite you to submit a revised version of the manuscript that addresses the points raised during the review process.

Also considering Covid-19 global alert, I would appreciate receiving your revised manuscript by May 31, 2020. To enhance the reproducibility of your results, we recommend that if applicable you deposit your laboratory protocols in protocols.io, where a protocol can be assigned its own identifier (DOI) such that it can be cited independently in the future. For instructions see: http://journals.plos.org/plosone/s/submission-guidelines#loc-laboratory-protocols

We look forward to receiving your revised manuscript.

Kind regards,

Francesca Megiorni, Ph.D.

Academic Editor

PLOS ONE

Journal Requirements:

2. We noticed minor instances of text overlap with the following previous publication(s), which need to be addressed:

(1) https://www.hindawi.com/journals/omcl/2018/7512159/

The text that needs to be addressed involves the second paragraph of the Introduction section.

In your revision please ensure you cite all your sources (including your own works), and quote or rephrase any duplicated text outside the methods section. Further consideration is dependent on these concerns being addressed."

3. To comply with PLOS ONE submission guidelines, in your Methods section, please provide additional information regarding your statistical analyses. For more information on PLOS ONE's expectations for statistical reporting, please see https://journals.plos.org/plosone/s/submission-guidelines.#loc-statistical-reporting.

**Comments to the Author**

1. Is the manuscript technically sound, and do the data support the conclusions?

Reviewer #1: Yes

2. Has the statistical analysis been performed appropriately and rigorously? 

Reviewer #1: Yes

3. Have the authors made all data underlying the findings in their manuscript fully available?

Reviewer #1: Yes

4. Is the manuscript presented in an intelligible fashion and written in standard English?

Reviewer #1: Yes

**5. Review Comments to the Author**

Reviewer #1: Thank you very much for inviting me to review the manuscript- “Unveiling Prognostics Biomarkers of Tyrosine Metabolism Reprogramming in Liver Cancer by Cross-platform Gene Expression Analyses” by Nguyen et al.

The authors evaluate the role of tyrosine metabolism in liver cancer development. They explore multiple public gene expression datasets and find that five Tyrosine catabolic enzymes are down regulated in HCC and also that they have prognostic significance.

The authors should include more details in the first three paragraphs of the results section like p values, number of samples, correlation ratios.

The TGCA data set has 50 normal tissues. The authors say that they use 160 normal tissues. Did they also include normal tissue from GTEX.

Did the stage of HCC in the TCGA cohort correlate with levels of the 5 enzymes.

It is not clear why the author chose to further explore GSTZ1 and not the other enzymes. The prognostic value of GSTZ1is not convincing based on the tcga data.

The authors show the pathway analysis of GSTZ1 overexpression in huh7 cells. They do not explain what the phenotype of overexpression was. Was it increased proliferation or decreased apoptosis?

The discussion section needs to be made more succinct and relevant to the current manuscript.

Figure 3 shows prognostic relevance of the five catabolic enzymes in the Tyrosine pathway. Four of the genes are divided by median and 182 patients are in the high and 182 pts in the low category. But GSTZ1 shows 73 patients in the high and 91 patients in the low. Can the author's explain why this is?

In figure 4 authors should include more representative samples from the public pathology database. They currently show just one core for each. Number of samples should be included in the y-axis

In Figure 6 authors show survival curves for mir539 and mir661. The number of patients in the denominator appear different for both of them. For mir539 the high category has 225 patients. For mir661 there are 82 patients in the high category.

I would skip this sentence “We further discovered that the expression of GSTZ1, the fourth rate limiting enzyme in tyrosine catabolism, regulates glycolytic gene expression”. Their analysis is exploratory and cannot be considered a true discovery.

Authors conclude by saying that they report a “novel function for tyrosine catabolic genes in tumorigenesis”. Other papers have alluded to a role for tyrosine metabolism in cancer. I would avoid using broad claims for novelty and focus on specific findings of this paper.

---

## [Author Response · Author response to Decision Letter 0]

13 May 2020

Legend: black (reviewer comment), blue (response), purple (text from manuscript).

Reviewers comments:

The authors evaluate the role of tyrosine metabolism in liver cancer development. They explore multiple public gene expression datasets and find that five Tyrosine catabolic enzymes are down regulated in HCC and also that they have prognostic significance.

We would like to thank the reviewer for the careful assessment of the manuscript and the thorough comments provided. The comments are fully addressed in the current letter and can be found in the revised manuscript.

Comment 1: The authors should include more details in the first three paragraphs of the results section like p values, number of samples, correlation ratios.

Response: Agree. We have added the information on p value, fold change. See page 5, line 8; page 6, line 2 and page 6, line 10.

Comment 2: The TGCA data set has 50 normal tissues. The authors say that they use 160 normal tissues. Did they also include normal tissue from GTEX.

Response: Yes, we included TCGA normal (adjacent tissue) and GTEx normal. It is stated in the method : “Normal tissues are matched TCGA adjacent tissue and GTEx normal tissue.”

Comment 3: Did the stage of HCC in the TCGA cohort correlate with levels of the 5 enzymes.

Response: To answer your question, we include the TCGA HCC stage plots for each genes. From stage I to II, TAT, HPD, HGD, GSTZ1, FAH levels decrease as the disease progresses. Except for GSTZ1, in other four genes, stage IV is an outlier where gene expression levels increased compared to earlier stages. 

Figure R1: Stage-wise gene expression profile of the tyrosine catabolic genes in HCC. Violin plots represent the gene expression level in terms of log2(TPM+1) in the tumor from Stage I through Stage IV. TPM, transcript per million.

Comment 4: It is not clear why the author chose to further explore GSTZ1 and not the other enzymes. The prognostic value of GSTZ1is not convincing based on the tcga data.

Response: This goes back to select the most important enzyme for further investigation. We made the decision to choose GSTZ1 based on three levels of data: (1) RNA level, (2) protein level and (3) prognosis. GSTZ1 decreases in tumor tissue compared to normal tissue at both RNA and protein levels. As for prognosis, HGD expression correlates stronger with survival than GSTZ1 (p = 0.0039 compared to p = 0.036). However, HGD expression does not significantly change at RNA level. So from our point of view, GSTZ1 is a better candidate among the two candidates.

Comment 5: The authors show the pathway analysis of GSTZ1 overexpression in huh7 cells. They do not explain what the phenotype of overexpression was. Was it increased proliferation or decreased apoptosis?

Response: Indeed. Thanks for pointing this out. We have added the explanations as to how increase in oxidative phosphorylation and other pathways might contribute to tumor suppression and apoptosis in liver cancer. Please see purple text at page 8, line 9 and line 13 in the manuscript.

“Noticeably, increased GSTZ1 expression led to heightened Oxidative Phosphorylation and Respiratory Electron Transport. Previously published in hepatocytes, limited oxidative phosphorylation activity associated with decreased apoptotic cell-death and increased cancer development. On the other hand, genes involved in glycolysis, such as HK2, PDK2 were downregulated (1.88-fold and 2.05-fold, respectively) in cell expressing GSTZ1 compared with vector control. It is known that the glycolytic gene HK2 is highly expressed in HCC. In cirrhosis, increased expression of glycolytic genes associated with higher HCC risk.” 

Comment 6: The discussion section needs to be made more succinct and relevant to the current manuscript.

Response: This has been rewritten where irrelevant information has been removed.

Comment 7: Figure 3 shows prognostic relevance of the five catabolic enzymes in the Tyrosine pathway. Four of the genes are divided by median and 182 patients are in the high and 182 pts in the low category. But GSTZ1 shows 73 patients in the high and 91 patients in the low. Can the author' s explain why this is?

Response: The reason for this difference is that we chose median as the cutoff value for other genes but for GSTZ1, we chose different optimal cutoff points. For this specific gene, we observe the difference in survival at this cutoff, rather than the median. Please see page 6, line 18 for explanation. To make it clearer, we have added more detail to the Method. Please see page 13, line 18-20.

“We set the high and low gene expression level groups by the median value for TAT, HPD HGD and FAH. For GSTZ1, the cutoff for high expression group is 75% and cutoff for low expression group is 25%.”

Comment 8: In figure 4 authors should include more representative samples from the public pathology database. They currently show just one core for each. Number of samples should be included in the y-axis

Response: Agree. We have added representative images of HPD, HGD and FAH as well as the number of samples in the y-axis to Figure 4. 

Comment 9: In Figure 6 authors show survival curves for mir539 and mir661. The number of patients in the denominator appear different for both of them. For mir539 the high category has 225 patients. For mir661 there are 82 patients in the high category.

Response: The reason for the difference in each category is meanwhile in HCC, the data for each microRNAs come from different data sets. There was no data set that contained survival data for both microRNAs. For miR-539, the corresponding dataset has 421 samples. On the other hand, for miR-661, the corresponding dataset has 166 samples. This was briefly mentioned in the Results. Thanks for pointing that out though since perhaps we have not made it clear at the Methods. We have added information about this on the Method section. Please see page 14, line 2-5.

Comment 10: I would skip this sentence “We further discovered that the expression of GSTZ1, the fourth rate limiting enzyme in tyrosine catabolism, regulates glycolytic gene expression”. Their analysis is exploratory and cannot be considered a true discovery.

Response: Agree. We have removed this sentence from the manuscript.

Comment 11: Authors conclude by saying that they report a “novel function for tyrosine catabolic genes in tumorigenesis”. Other papers have alluded to a role for tyrosine metabolism in cancer. I would avoid using broad claims for novelty and focus on specific findings of this paper.

Response: Agree. This conclusion has been removed to faithfully reflect our findings.

---

## [Editor Report · Decision Letter 1]

28 May 2020

Unveiling Prognostics Biomarkers of Tyrosine Metabolism Reprogramming in Liver Cancer by Cross-platform Gene Expression Analyses

PONE-D-20-02954R1

Dear Dr. Nguyen,

I carefully checked your comments and addictions to your paper and I am pleased to inform you that your manuscript has been now judged scientifically suitable for publication and will be formally accepted for publication once it complies with all outstanding technical requirements.

With kind regards,

Francesca Megiorni, Ph.D.

Academic Editor

PLOS ONE

---

## [Editor Report · Acceptance letter]

2 Jun 2020

PONE-D-20-02954R1 

Unveiling Prognostics Biomarkers of Tyrosine Metabolism Reprogramming in Liver Cancer by Cross-platform Gene Expression Analyses 

Dear Dr. Nguyen:

I'm pleased to inform you that your manuscript has been deemed suitable for publication in PLOS ONE. Congratulations! Your manuscript is now with our production department. 

Kind regards, 

on behalf of

Dr. Francesca Megiorni 

Academic Editor

PLOS ONE